# Heterologous ChAdOx1 nCoV-19/BNT162b2 Prime-Boost Vaccination Induces Strong Humoral Responses among Health Care Workers

**DOI:** 10.3390/vaccines9080857

**Published:** 2021-08-04

**Authors:** Louise Benning, Maximilian Töllner, Asa Hidmark, Matthias Schaier, Christian Nusshag, Florian Kälble, Paula Reichel, Mirabel Buylaert, Julia Grenz, Gerald Ponath, Katrin Klein, Martin Zeier, Caner Süsal, Paul Schnitzler, Christian Morath, Claudius Speer

**Affiliations:** 1Department of Nephrology, University Hospital Heidelberg, 69120 Heidelberg, Germany; maximilian.toellner@googlemail.com (M.T.); asa.hidmark@gmail.com (A.H.); matthias.schaier@med.uni-heidelberg.de (M.S.); christian.nusshag@med.uni-heidelberg.de (C.N.); florian.kaelble@med.uni-heidelberg.de (F.K.); paulareichel@web.de (P.R.); mirabel.buylaert@med.uni-heidelberg.de (M.B.); julia.grenz@med.uni-heidelberg.de (J.G.); gerald.ponath@gmail.com (G.P.); katrin.klein@med.uni-heidelberg.de (K.K.); Martin.Zeier@med.uni-heidelberg.de (M.Z.); christian.morath@med.uni-heidelberg.de (C.M.); 2Institute of Immunology, University Hospital Heidelberg, 69120 Heidelberg, Germany; caner.suesal@med.uni-heidelberg.de; 3Department of Infectious Diseases, Virology, University Hospital Heidelberg, 69120 Heidelberg, Germany; paul.schnitzler@med.uni-heidelberg.de; 4Molecular Medicine Partnership Unit Heidelberg, EMBL, 69117 Heidelberg, Germany

**Keywords:** COVID-19, COVID-19 vaccination, heterologous vaccination, humoral response, mRNA vaccine, BNT162b2, adenovirus-vectored vaccine, ChAdOx1 nCoV-19, SARS-CoV-2

## Abstract

Despite limited data on safety and immunogenicity, heterologous prime-boost vaccination is currently recommended for individuals with ChAdOx1 nCoV-19 prime immunization in certain age groups. In this prospective, single-center study we included 166 health care workers from Heidelberg University Hospital who received either heterologous ChAdOx1 nCoV-19/BNT162b2, homologous BNT162b2 or homologous ChAdOx1 nCoV-19 vaccination between December 2020 and May 2021. We measured anti-S1 IgG, SARS-CoV-2 specific neutralizing antibodies, and antibodies against different SARS-CoV-2 fragments 0–3 days before and 19–21 days after boost vaccination. Before boost, 55/70 (79%) ChAdOx1 nCoV-19-primed compared with 44/45 (98%) BNT162b2-primed individuals showed positive anti-S1 IgG with a median (IQR) anti-S1 IgG index of 1.95 (1.05–2.99) compared to 9.38 (6.26–17.12). SARS-CoV-2 neutralizing antibodies exceeded the threshold in 24/70 (34%) of ChAdOx1 nCoV-19-primed and 43/45 (96%) of BNT162b2-primed individuals. After boosting dose, median (IQR) anti-S1 IgG index in heterologous ChAdOx1 nCoV-19/BNT162b2 vaccinees was 116.2 (61.84–170), compared to 13.09 (7.03–29.02) in homologous ChAdOx1 nCoV-19 and 145.5 (100–291.1) in homologous BNT162b2 vaccinees. All boosted vaccinees exceeded the threshold for neutralization, irrespective of their vaccination scheme. Vaccination was well-tolerated overall. We show that heterologous ChAdOx1 nCoV-19/BNT162b2 vaccination is safe and induces a strong and broad humoral response in healthy individuals.

## 1. Introduction

Safety and efficacy of the BNT162b2 mRNA COVID-19 vaccine as well as of the chimpanzee adenovirus-vectored vaccine ChAdOx1 nCoV-19 have been described in great detail for homologous prime-boost vaccination [1,2]. Initially, the incidence of serious adverse events was deemed low and described as similar in the vaccine and placebo groups or not related to study vaccine. As vaccine rollout began, new reports of adverse events emerged, especially rare cases of thrombotic thrombocytopenia after ChAdOx1 nCoV-19 vaccination due to the formation of pathologic antibodies to platelet factor 4 [3,4,5]. As a consequence, several countries adapted their vaccination strategy, preferably vaccinating the elderly with ChAdOx1 nCov-19 as severe adverse events were hardly reported in this age group. In Germany, individuals younger than 60 years who were previously vaccinated with ChAdOx1 nCoV-19 have now been recommended to receive BNT162b2 as a booster dose [6]. Deviation from recommendation is possible after individual risk analysis and sufficient patient information. Lately, Shaw et al. demonstrated an increase in systemic reactogenicity after the boost dose in participants who received heterologous prime-boost vaccination compared to participants receiving homologous BNT162b2 or ChAdOx1 nCoV-19 vaccination [7]. Given the scarcity of data regarding the immunogenicity of mRNA boosting after primary ChAdOx1 nCoV-19 vaccination, there is an urgent need to determine humoral responses following heterologous vaccination. We investigated anti-S1 IgG and SARS-CoV-2 specific neutralizing antibodies in 166 health workers from the Heidelberg University Hospital, Germany, and compared reactogenicity and immunogenicity after vaccination in either homologous BNT162b2, homologous ChAdOx1 nCoV-19 or heterologous ChAdOx1 nCoV-19/BNT162b2 vaccinated individuals.

## 2. Materials and Methods

### 2.1. Study Design and Cohorts

In this prospective, single-center, observational cohort study, 166 health care workers with different COVID-19 vaccination schemes between December 2020 and June 2021 at the Department of Nephrology of the University Hospital of Heidelberg were included. As a priming dose, 84 participants received the chimpanzee adenovirus-vectored vaccine ChAdOx1 nCoV-19 (AstraZeneca, AZ, Cambridge, United Kingdom), and 82 participants received the BNT162b2 mRNA vaccine (BioNTech, BNT, Mainz, Germany), whereof serum for analysis was obtained in 70 AZ-primed and 45 BNT-primed participants. In previously AZ-primed participants, 17 health care workers received AZ as a homologous boosting dose, and 35 participants received BNT as a heterologous boosting dose. All BNT-primed participants were also boosted with BNT (Figure 1). The immunogenicity of different vaccination regimens was determined as described in detail below.

The median (interquartile range, IQR) prime-boost interval was 83 (77–84) days for heterologous AZ/BNT vaccinees, 20 (20–20) days for homologous BNT/BNT vaccinees, and 82 (82–83) days for homologous AZ/AZ vaccinees, respectively.

Antibodies to the nucleocapsid protein were measured before prime as well as boost vaccination. Individuals with positivity were excluded because of suspected recent SARS-CoV-2 infection.

Anti-S1 IgG index and SARS-CoV-2 specific neutralizing antibodies were measured 0–3 days before and after a median (IQR) of 20 (19–21) days after boost vaccination. In addition, a multiplex bead-based assay was performed in 30 age- and sex-matched participants after heterologous AZ/BNT or homologous BNT/BNT vaccination to detect different SARS-CoV-2 target antibodies before and after the boosting dose.

The study is part of an ongoing single-center study to determine immunogenicity of different COVID-19 vaccines in renal transplant recipients compared to health care workers (DRKS00024668) and was approved by the ethics committee of the University of Heidelberg and conducted in accordance with the Declaration of Helsinki. Written informed consent was obtained from all study participants.

### 2.2. Anti-SARS-CoV-2 IgG Chemiluminescent Immunoassay

SARS-CoV-2 Total Assay (Siemens, Eschborn, Germany) was used to determine IgG response against the S1 protein. The result is expressed as a dimensionless index, a semi-quantitative index <1 was classified as negative, a value of ≥1 as positive. The cut-off value was set three standard deviations above the mean of the negatives, according to the manufacturer. The defined cutoff gives a specificity of 100% and a sensitivity of 89% for the detection of anti-S1 antibodies.

### 2.3. Detection of SARS-CoV-2 Neutralizing Antibodies

A plate-based SARS-CoV-2 surrogate virus neutralizing assay (Medac, Wedel, Germany) was used to identify the binding-inhibition potency of serum samples as previously described [8]. The test is based on antibody-mediated blockade of the interaction between the angiotensin-converting enzyme 2 (ACE2) receptor protein and the receptor-binding domain (RBD) of the SARS-CoV-2 spike protein and mimics the virus–host interaction by direct protein–protein interaction. A cut-off of ≥30% inhibition of receptor-binding domain–ACE-2 binding was applied according to the manufacturer’s instructions. The test achieves 99.93% specificity and 95–100% sensitivity.

### 2.4. Bead-Based Multiplex Assay for SARS-CoV-2 Antibody Detection

To identify IgG antibodies against different SARS-CoV-2 target antigens, a multiplex bead-based assay for the Luminex platform (LabScreen COVID Plus) was performed (One Lambda Inc., West Hill, CA, USA) [9]. The assay detects participant’s antibody response to 5 different SARS-CoV-2 proteins, namely, the full spike protein, 3 individual domains of the spike protein (S1, S2, and receptor binding domain), and the nucleocapsid. In addition, antibody reactivity against 6 other coronaviruses is also detected, namely HCoV-229E, HCoV-HKU1, HCoV-NL63, HCoV-OC43, MERS-CoV, and SARS-CoV-1. Antibody detection was performed as described previously and the mean fluorescence intensity (MFI) was analyzed on a Luminex 200 device (Luminex Corporation, Noord-Brabant, The Netherlands). The cutoff values for each recombinant protein are given in Appendix A.

### 2.5. Monitoring of Adverse Events

Adverse events were assessed using a 12-item questionnaire inquiring previously mentioned side-effects after vaccination and the use of pain medication after vaccine reception. The content of the questionnaire is shown in the Appendix A. Local events included pain at the injection site, redness and swelling. Systemic events included fever ≥38 °C, chills, headache, joint pain, muscle pain, vomiting, diarrhea, swollen lymph nodes, and fatigue.

### 2.6. Statistics

Data are expressed as median and IQR or number (N) and percent (%). When the impact of the priming dose was analyzed, different groups were compared using the Mann–Whitney *U* test in the case of continuous variables and Fisher’s exact test in the case of categorial variables. For the analysis of the impact of the boosting dose, results of the three different groups were compared by applying the Kruskal–Wallis test with Dunn’s post-test. Statistical analysis of categorical data was performed using the chi-square (χ^2^) test. For the analysis of the impact of the priming and boosting dose in the same participant, Wilcoxon rank sum test was applied. Statistical significance was assumed at a *p*-value < 0.05. The statistical analysis was performed using GraphPad Prism version 9.0.0 (GraphPad Software, San Diego, CA, USA).

## 3. Results

### 3.1. Baseline Characteristics

From 29 December 2020 to 1 June 2021, we prospectively enrolled 166 health care workers who had either AZ or BNT as the priming dose, homologous AZ/AZ or BNT/BNT combinations as the boosting dose, or heterologous BNT boost after AZ as the priming dose (AZ/BNT).

For priming dose, 115 health care workers were analyzed, who received either AZ (N = 70) or BNT (N = 45). Median (IQR) age at enrollment was 36 (27–55) years for participants receiving AZ and 45 (33–55) years for participants receiving BNT. Participants receiving BNT for priming dose were significantly older (*p* = 0.03; Table 1). Forty-seven (67%) participants with AZ priming dose and 32 (71%) participants with BNT priming dose were female. Baseline characteristics are shown in Table 1.

For boosting dose, 17 participants received homologous AZ boosting, 35 received heterologous BNT boosting after AZ priming, and 82 received homologous BNT boosting. Median (IQR) age for participants receiving homologous AZ/AZ boosting was 55 (33–60) years, for participants receiving heterologous AZ/BNT vaccination 30 (24–45) years, and for participants receiving homologous BNT/BNT boosting 45 (33–56) years (Table 1). Participants receiving heterologous AZ/BNT were significantly younger than participants receiving homologous AZ/AZ or BNT/BNT (for both *p* < 0.001, Table 1). Eleven (65%) of the homologous AZ/AZ vaccinees, 23 (66%) of the heterologous AZ/BNT vaccinees, and 63 (77%) of the homologous BNT/BNT were female, respectively. No statistical significance regarding sex was observed between homologous AZ/AZ, heterologous AZ/BNT or homologous BNT/BNT vaccinees (*p* = 0.35).

### 3.2. Heterologous AZ/BNT Boost Elicits Strong Anti-S1 IgG Antibody Response with High Neutralization Capacity

After AZ or BNT priming dose, 55/70 (79%) and 44/45 (98%) vaccinees showed anti-S1 IgG levels above the threshold. AZ-primed individuals had with a median of 1.9 (IQR 1.0–3.0) a significantly lower anti-S1 IgG level compared to the 9.4 level (6.3–17.1) in BNT-primed individuals (*p* < 0.001; Figure 2A). Antibodies of 24/70 (34%) AZ- and 43/45 (96%) BNT-primed individuals exceeded the threshold for neutralization (*p* < 0.001). Neutralizing antibody capacity was significantly lower in AZ-primed individuals compared to BNT-primed individuals with a median (IQR) percent inhibition of 15.7 (4.8–39.4) compared to 68.7 (50.9–75.4) (*p* < 0.001, Figure 2B).

After the boosting dose, all individuals exceeded the threshold for the anti-S1 IgG level, including the individuals who were below the cutoff after the priming dose (Figure 2A). Heterologous AZ/BNT vaccinees had comparable anti-S1 IgG levels to homologous BNT/BNT vaccinees with median (IQR) anti-S1 IgG indices of 116.2 (61.8–170) compared to 145.5 (100.0–291.1). With a median (IQR) anti-S1 IgG level of 13.1 (7.0–29.0), homologous AZ/AZ-boosted individuals showed significantly lower levels, as compared to heterologous AZ/BNT-boosted individuals or homologous BNT/BNT-boosted individuals (for both *p* < 0.001; Figure 2A). Antibodies of all individuals exceeded the threshold for neutralization after the respective boosting dose. Median (IQR) percent inhibition was 93.5 (88.6–96.7) for homologous AZ/AZ-boosted vaccinees, 96.8 (96.7–96.9) for heterologous AZ/BNT-boosted vaccinees, and 97.0 (96.1–98.0) for homologous BNT/BNT-boosted vaccinees, respectively (Figure 2B). No statistically significant difference was observed between homologous BNT/BNT-boosted and heterologous AZ/BNT vaccinated individuals. Homologous AZ/AZ vaccinees showed a significantly lower neutralizing antibody capacity compared to heterologous AZ/BNT and homologous BNT/BNT-boosted vaccinees (*p* = 0.001 and *p* < 0.001, Figure 2B).

For 16 homologous AZ/AZ, 22 heterologous AZ/BNT, and 45 homologous BNT/BNT prime-boost vaccinees, paired anti-S1 IgG and neutralizing antibody courses after the first and second vaccination in every participant are shown in Figure 2C. When the individual course after the priming and boosting dose was compared, all individuals significantly improved in neutralizing antibody capacity, even those individuals who were below the threshold of neutralization after the priming dose (Figure 2C).

### 3.3. Broad Humoral Response against Different SARS-CoV-2 Spike Fragments after Heterologous AZ/BNT and Homologous BNT/BNT Boosting Dose

In addition, we determined the IgG reactivity against 4 different fragments of the SARS-CoV-2 spike-protein as well as the IgG reactivity against the SARS-CoV-2 nucleocapsid protein in 30 age- and sex-matched heterologous AZ/BNT (N = 15) and homologous BNT/BNT (N = 15) vaccinees before and after boosting dose. Participants’ baseline characteristics are given in Appendix A.

After the first dose, BNT-primed vaccinees showed higher reactivity against different SARS-CoV-2 target antigens compared to AZ-primed individuals. After the priming dose, 12/15 (80%) of AZ-primed, compared to 14/15 (93%) of BNT-primed, individuals had MFI values above the cutoff for detection for the full spike and the S1 protein (*p* = 0.6 and *p* = 0.6). The threshold for reactivity against the RBD protein was exceeded in 9/15 (60%) of AZ-primed, compared to 14/15 (93%) of BNT-primed, individuals (*p* = 0.08). With 9/15 (60%) BNT-primed individuals compared to 1/15 (7%) AZ-primed individuals, significantly more BNT-primed individuals exceeded the threshold for reactivity against the S2 protein (*p* = 0.005). BNT-primed vaccinees showed significantly higher MFI values than AZ-primed vaccinees against the full spike protein (21,729 (19,995–2340) vs. 14,909 (10,589–16,111); *p* < 0.001), the S1 protein (7707 (6504–10,001) vs. 4968 (3183–5648); *p* < 0.001), the RBD protein (11,929 (9607–14,173) vs. 6505 (4030–8640); *p* < 0.001), and the S2 protein (6043 (1381–7970) vs. 1262 (552–1899); *p* = 0.002). No AZ- or BNT-primed individual showed reactivity above the cutoff against the nucleocapsid protein, confirming the exclusion of previously infected participants. Antibodies against the full spike protein, the S1 spike protein, the RBD protein, the S2 spike protein, and the nucleocapsid protein are shown in Figure 3A–E.

After the boosting dose, all 15 heterologous AZ/BNT (100%) and all 15 homologous BNT/BNT (100%) vaccinees exceeded the cutoff for detection for the full spike protein, the S1 protein, and the RBD protein. For the S2 protein, 14/15 (93%) of heterologous AZ/BNT-boosted vaccinees and 15/15 (100%) of homologous BNT-boosted vaccinees showed reactivity above the cutoff. MFI values for the full spike protein (24,243 (24,066–24,335) vs. 23,849 (22,692–24,008); *p* < 0.001), the S1 protein (19,332 (18,268–20,308) vs. 16,955 (14,708–18,420); *p* = 0.003), and S2 protein (13,138 (11,364–18,743) vs. 9696 (6369–12,556); *p* = 0.004) were significantly higher in heterologous AZ/BNT-boosted compared to homologous BNT/BNT-boosted individuals, respectively (Figure 3A,B,D). No differences were observed in MFI values against the RBD protein or the nucleocapsid protein (Figure 3C,E). Figure 3F–J show the paired individual prime to boost effect regarding IgG antibodies against different SARS-CoV-2 target antigens.

Regarding reactivity against other coronaviruses, statistical significance was found in BNT-primed individuals compared to AZ-primed individuals for HCoV-NL-63 (*p* = 0.041). Heterologous AZ/BNT-boosted individuals showed significantly higher reactivity against the S1 spike protein of SARS-CoV-1 with a median (IQR) MFI of 3708 (2934–3980) compared to 1293 (779–1649) in homologous BNT-boosted individuals (*p* < 0.001). No other statistical differences were found between the groups in IgG reactivities against other coronaviruses (Appendix A).

### 3.4. Local and Systemic Reactions after Different Vaccination Regimens

Local and systemic reactions were assessed in the study population using a 12-item questionnaire and information regarding reactogenicity was obtained in 50 AZ-primed, 53 BNT-primed, 14 homologous AZ/AZ-, 29 heterologous AZ/BNT-, and 53 homologous BNT/BNT-boosted individuals. Any reaction, either local or systemic, was most commonly reported by AZ-primed individuals, followed, in decreasing order, by homologous BNT/BNT, heterologous AZ/BNT, BNT-primed, and homologous AZ/AZ vaccinees (92% compared to 83%, 72%, 60%, 29%; *p* < 0.001, Figure 4).

After priming dose, local reactions, such as redness, pain, and tenderness at injection site, were observed in 25/50 (50%) of AZ-primed individuals compared to 26/53 (49%) of BNT-primed individuals with no statistical significance. Systemic reactions after priming dose, such as fever, fatigue, headache, chills, muscle or joint pain, were significantly more often reported in AZ-primed individuals compared to BNT-primed individuals (86% to 26%, respectively; *p* < 0.001).

When comparing reactogenicity in homologous AZ/AZ to heterologous AZ/BNT and homologous BNT/BNT vaccinees after boosting dose, local reactions were significantly lower in homologous AZ/AZ-boosted individuals (7% compared to 52% and 53%, respectively; *p* = 0.0071). Systemic reactions were highest in homologous BNT/BNT-boosted individuals compared to heterologous AZ/BNT- and homologous AZ/AZ-boosted vaccinees (76% compared to 52% and 29%, respectively; *p* = 0.0025).

## 4. Discussion

A recent correspondence by Shaw et al. on interim results of the Com-COV-study showed higher reactogenicity in heterologous prime-boost vaccinees compared to participants with homologous vaccination regimens [7]. However, peer-reviewed immunogenicity data for heterologous prime-boost vaccination remains scarce. This is one of the first studies to provide an in-depth examination of humoral responses after heterologous prime-boost vaccination. We show that heterologous AZ/BNT prime-boost vaccination induces a strong and broad immunogenicity against various SARS-CoV-2 spike protein fragments with high neutralizing antibody capacity.

Borobia et al. recently published a pre-print on a phase-II trial of 676 individuals randomized either to observation after AZ priming or intervention in the form of heterologous BNT boosting (CombiVac S) [10]. IgG antibodies against the SARS-CoV-2 spike protein, SARS-CoV-2 neutralizing antibodies, and cellular immune response significantly increased in BNT-boosted individuals, compared to only AZ-primed individuals [10]. These findings are confirmed by our data, as we also demonstrate significantly higher anti-S1 IgG levels as well as the neutralizing capacity of SARS-CoV-2-specific antibodies in heterologous AZ/BNT vaccinated individuals compared to homologous AZ/AZ vaccinees. However, no comparison to homologous BNT/BNT vaccinated participants was made in the study by Borobia et al. [10]. We demonstrate that heterologous AZ/BNT vaccinees have at least comparable humoral responses to homologous BNT/BNT vaccinated individuals. In a detailed bead-based analysis of different SARS-CoV-2 target antigens, heterologous AZ/BNT vaccinees have higher levels of antibodies against the full spike, the S1 spike, and the S2 spike protein, whereas antibodies against the RBD, including the neutralizing antibody capacity, were comparable between both groups. These results are in accordance with recently pre-print published immunogenicity data by Liu et al. from the Com-COV Study Group and immunogenicity data by Hillus et al. [11,12]. Liu et al. investigated the induction of humoral and cellular responses in a large cohort of 463 individuals receiving different prime-boost vaccination regimens. They demonstrated the non-inferiority of a heterologous AZ/BNT vaccination regimen compared to a homologous AZ/AZ vaccination scheme regarding both the induction of humoral and cellular response after boost vaccination. As the trial was designed as a non-inferiority trial, no claim was made regarding the superiority of a vaccination scheme; however, the geometric mean concentration of antibodies was shown to be significantly higher in heterologous AZ/BNT-boosted individuals than that of homologous AZ/AZ vaccinees [11]. Hillus et al. also investigated humoral and cellular responses to different SARS-CoV-2 vaccination regimens in 340 health care workers after prime and boost immunization with different vaccination regimens [12]. Comparable to our own data, they demonstrate a strong increase in antibody levels in heterologous AZ/BNT vaccinees with initially low antibody response after AZ priming.

Notably, in our study, the BNT boost also elicits a strong and broad humoral response in individuals with antibody levels below the cutoff after AZ priming. Although antibody levels were not detectable in these participants, the SARS-CoV-2 specific cellular immune response (e.g., T-helper cells and memory B cells) may have been stimulated by the AZ priming dose, resulting in the strong induction of antibodies with a single BNT boost. Patients previously infected with SARS-CoV-2 with low antibody levels but robust cellular immune responses also showed strong antibody induction after receiving a single mRNA vaccination 6 months after infection, further supporting our hypothesis of primed cellular immunity [13,14,15]. However, because our study does not include cellular immunity data, we cannot map the complex vaccine-induced immunologic response and potential protection after AZ or BNT priming dose in individuals below the cutoff for humoral immunity.

We further show that antibody levels are significantly lower in AZ-primed compared to BNT-primed vaccines and in homologous AZ/AZ-boosted vaccines compared to homologous BNT/BNT- or heterologous AZ/BNT-boosted vaccinees. However, these results should be interpreted with caution. All AZ-boosted individuals meet the respective threshold for detection or neutralization and there are still no universally validated and accepted antibody cutoffs that correlate with protection against severe COVID-19 courses. In addition, data from the UK COVID-19 infection survey suggest a reduction in SARS-CoV-2 infection after only single-dose vaccination, with no statistical difference in those primed with BNT or AZ [16].

With emerging variants of concern, such as B.1.1.7 (alpha), B.1.351 (beta), B.1.617.2 (delta) and P.1 (gamma), with B.1.617.2 being the most dominant SARS-CoV-2 variant in Germany today, the effectiveness of different vaccination schemes also needs to be determined with respect to the neutralization of these VOC. Groß et al. and Barros-Martins et al. recently reported in small cohorts with different vaccination schemes, including 55 and 26 heterologous AZ/BNT vaccinees, high titers of neutralizing antibodies against the B.1.1.7, B.1.351 and P.1 VOC. Both groups report on particularly strong neutralization against VOCs in individuals who received a heterologous AZ/BNT compared to homologous AZ/AZ vaccination scheme [17,18]. As their results on spike-specific IgG are in line with our study results, we can assume an equally good neutralization against VOCs in our study population.

Regarding reactogenicity, we show that systemic responses were highest in individuals after AZ priming dose and after homologous BNT/BNT boosting dose. Homologous AZ/AZ-boosted individuals reported significantly fewer reactions after boosting than heterologous AZ/BNT- or homologous BNT/BNT-boosted participants, which is consistent with results recently reported by Shaw et al. and Hillus et al. [7,12]. The interim results of the Com-COV-trial revealed higher reactogenicity in heterologous AZ/BNT compared to homologous BNT/BNT-boosted vaccinees, which we do not see in our study. However, the time interval between the first (AZ) and second (BNT) vaccine dose was only 28 days in the Com-COV-trial compared to 83 days in our study, which may have influenced reactogenicity.

Limitations of our studies are the relatively small study size and the female predominance of our study population. However, neither Shaw et al. nor Hillus et al. reported on significant differences regarding the reactogenicity of different vaccination schemes in males or females [7,12]. We performed no sex-based analysis on reactogenicity due to the female predominance and the absence of any severe adverse events in our study population. Regarding immunogenicity, all individuals of our study population exceed the respective threshold for detection of anti-S1 IgG and neutralizing antibody capacity after boosting dose, regardless of their sex. Further studies should include a stratification for sex for evaluation of both, reactogenicity and immunogenicity, as immunity is known to be sex-dependent [19].

Other confounders affecting immune response are age and comorbidity. Since we investigated humoral responses following SARS-CoV-2 vaccination in health care workers, most individuals were between 25–65 years of age and in a comparably good general state of health. Since we, and others, have shown significantly reduced immunogenicity of SARS-CoV-2 vaccines in immunosuppressed patients, patients on dialysis, or even elderly individuals, our results cannot be transferred unrestricted to the general population [20,21,22]. Results have to be separately confirmed by including these vulnerable subpopulations into further studies. However, because health care workers are at particularly high risk for SARS-CoV-2 infection, there is a great need to determine safety and the immunogenicity of different COVID-19 vaccination regimens in this population.

This is one of the first studies to investigate in-depth humoral responses after heterologous AZ/BNT, compared to homologous BNT/BNT or AZ/AZ vaccination. We show that heterologous AZ/BNT vaccination seems to be safe and induces a strong and broad humoral response against various SARS-CoV-2 spike protein fragments with high neutralizing antibody capacity in healthy individuals. Further studies to investigate the efficacy of heterologous vaccination strategies against emerging variants of concern or in vulnerable subpopulations with reduced vaccine response are urgently needed.

## Figures and Tables

**Figure 1 vaccines-09-00857-f001:**
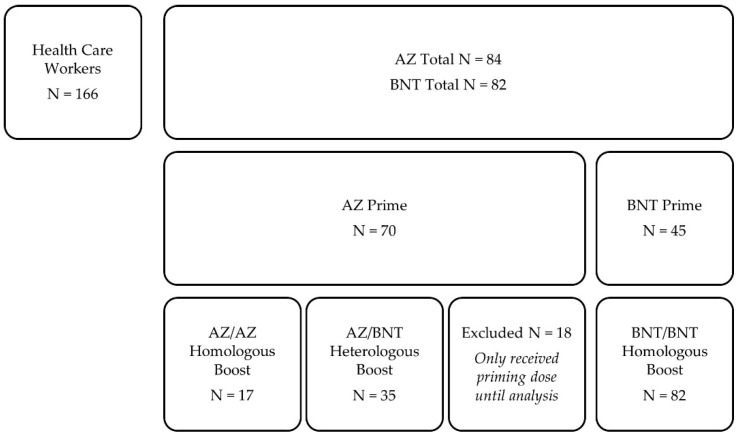
Flow chart on participants’ vaccination patterns. AZ, AstraZeneca; BNT, BioNTech; N, number.

**Figure 2 vaccines-09-00857-f002:**
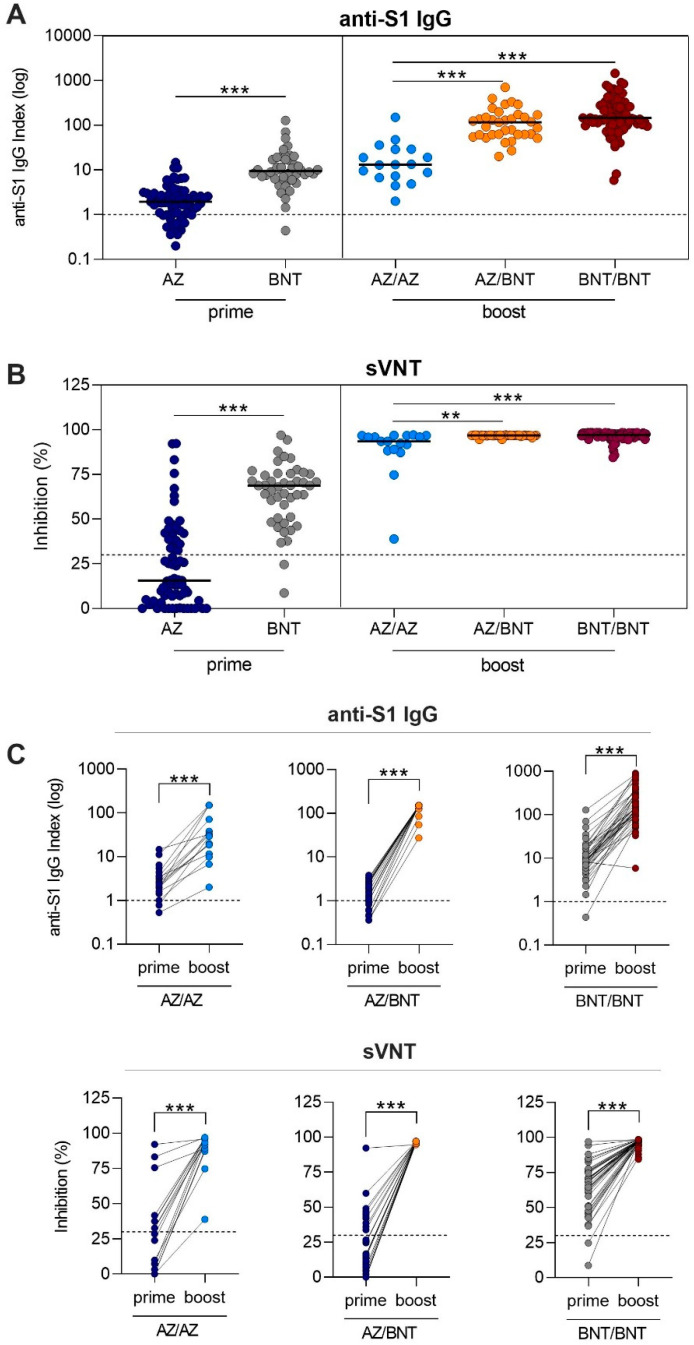
Anti-S1 IgG and neutralizing antibody response in health care workers after different SARS-CoV-2 prime-boost vaccination regimens. SARS-CoV-2 IgG antibodies represented, logarithmically, as an anti-S1 IgG index in health care workers after AZ and BNT prime vaccination and after AZ/AZ homologous, AZ/BNT heterologous, and BNT/BNT homologous boost vaccination (**A**). The dashed black line represents the cutoff for detection. A semi-quantitative index of <1 was classified as negative. Neutralizing antibody capacity measured by a surrogate virus neutralization test after AZ and BNT prime vaccination and after AZ/AZ homologous, AZ/BNT heterologous, and BNT/BNT homologous boost vaccination (**B**). The dashed black line represents the cut off for viral neutralization in this assay according to the manufacturer’s instructions. A cut off of <30% binding inhibition indicates absence or a level of SARS-CoV-2 neutralizing antibodies below the limit of detection of this test. Paired anti-S1 IgG and neutralizing antibody courses after the first and second vaccination (**C**). ** *p* < 0.01; *** *p* < 0.001.

**Figure 3 vaccines-09-00857-f003:**
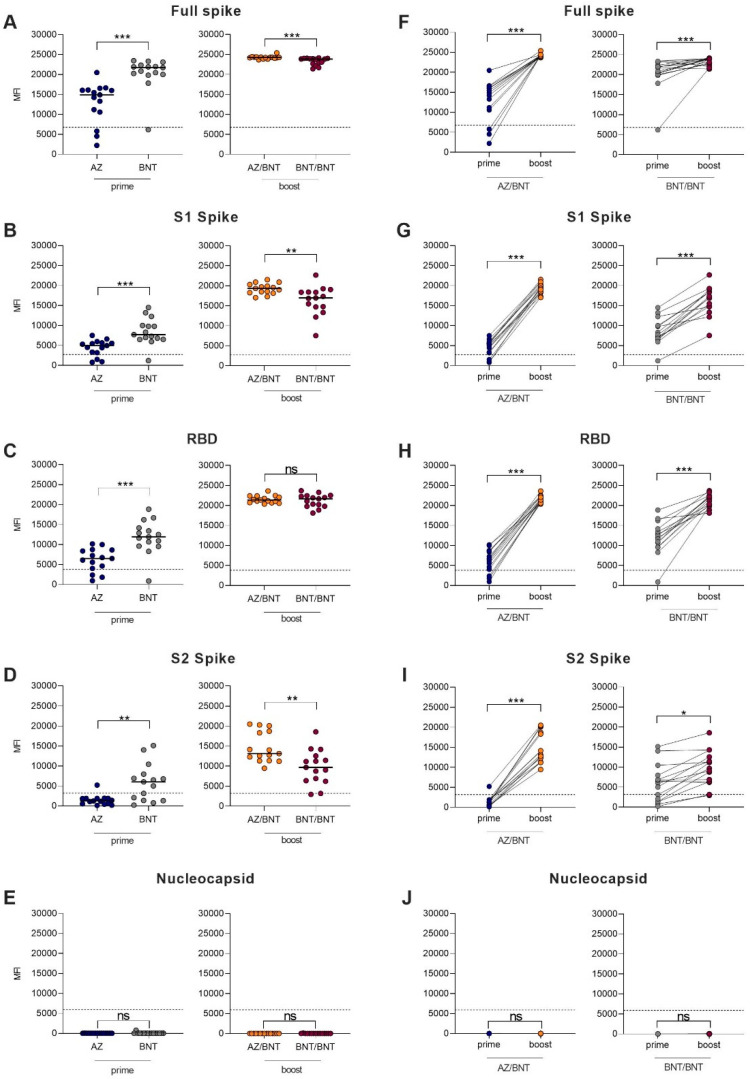
IgG antibodies against different SARS-CoV-2 target antigens in health care workers after AZ and BNT prime vaccination and after AZ/BNT heterologous and BNT/BNT homologous boost vaccination. Detection of antibodies against the full spike protein (**A**), the S1 spike protein (**B**), the receptor-binding domain of the spike protein (**C**), the S2 spike protein (**D**), and the nucleocapsid protein (**E**) of SARS-CoV-2 in health care workers after AZ and BNT prime vaccination and after AZ/BNT heterologous and BNT/BNT homologous boost vaccination. The *x*-axis represents the vaccination pattern, and the *y*-axis the mean fluorescence intensity value of the reactivity. The dashed black line represents the cutoff for detection for each target, respectively. The paired detection of antibodies against the full spike protein (**F**), the S1 spike protein (**G**), the receptor-binding domain of the spike protein (**H**), the S2 spike protein (**I**), and the nucleocapsid protein (**J**) after the first and second vaccination in health care workers with either AZ/BNT heterologous and BNT/BNT homologous boost vaccination. The *x*-axis represents priming and boosting dose and the *y*-axis the mean fluorescence intensity value of reactivity. * *p* < 0.05; ** *p* < 0.01; *** *p* < 0.001.

**Figure 4 vaccines-09-00857-f004:**
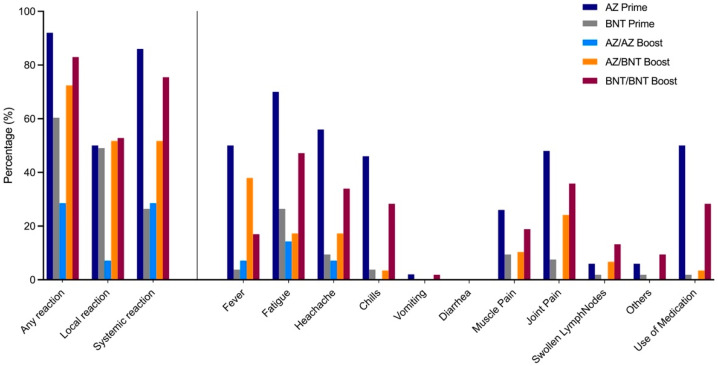
Local and systemic reactions in health care workers after AZ or BNT priming dose or homologous AZ/AZ, heterologous AZ/BNT, or homologous BNT/BNT vaccination.

**Table 1 vaccines-09-00857-t001:** Participants’ baseline characteristics. AZ, AstraZeneca; BNT, BioNTech; IQR, interquartile range; N, number.

	Total AZ		Total BNT	*p*-Value
N	84		82	
Age, median (IQR)	36 (26–55)		45 (33–56)	0.003
Female, N (%)	56 (67)		63 (77)	0.17
**Priming Dose**				
Vaccine	**AZ**		**BNT**	
N	70		45	
Age, median (IQR)	36 (27–55)		45 (33–55)	0.03
Female, N (%)	47 (67)		32 (71)	0.69
**Boosting Dose**				
Vaccine	**Homologous AZ/AZ**	**Heterologous AZ/BNT**	**Homologous BNT/BNT**	
N	17	35	82	
Age, median (IQR)	55 (33–60)	30 (24–45)	45 (33–56)	<0.001
Female, N (%)	11 (65)	23 (66)	63 (77)	0.35

## Data Availability

The data of this study are available on request from the corresponding authors.

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
