# Peer review of "Heterologous ChAdOx1 nCoV-19/BNT162b2 Prime-Boost Vaccination Induces Strong Humoral Responses among Health Care Workers"

_vaccines, 2021, doi:10.3390/vaccines9080857_

Round 1
Reviewer 1 Report
A preprint of similar study is on the Lancet preprint server [Liu et al], is in press and needs to be referenced. It had more participants and examined T cell responses. The antibody responses to spike protein were however similar and such independent studies with similar findings contributes to confidence in the results.
Author Response
Thank you for your comments. We already included data on reactogenicity published by Shaw et al. from the Com-COV Study Group but now, as reactogenicity data is available from the same trial, we will also include a reference to Liu et al. in our discussion.
Reviewer 2 Report
This paper deals with the boosting effect of the same or different vaccines against SARS-CoV-2, and the results are of great interest. It contains important findings for COVID-19, and the data may be sufficient for publication.
Please improve on the points requested below.
- Please improve the position of the items and vertical lines in Table 1. In particular, “N”, “Age” is corresponding to “Item” not “Total AZ”.
“Total AZ” should be listed above “84”.
- In Figure 2 and 3: The presence or absence of significant differences cannot be determined without expansion. please make the display of significant differences a little larger.
- In Discussion section: The differences between two vaccine antigens are full length of spike protein (BNT) and glycoprotein in spike protein (AZ). Nevertheless, AZ/BNT combination showed higher antibody titer than homologous BNT at S1 and S2. Can you reflect on your speculations about this phenomenon?
- In Discussion section (lines 374-388): Were there any differences in adverse reactions between female and male? Please add a special note in this paragraph.
- In Discussion section: If you provide a citation, please add the reference number to the text concerned (lines 315, 323, 332, 368…).
Author Response
Thank you for your valuable comments and suggestions.
Regarding 1)
We repositioned the items in Table 1 and adjusted the formatting of the table accordingly. "Total AZ" is now listed above "84".
Regarding 2)
We further adjusted the display of significant differences accordingly (font size 16 compared to size 8 beforehand).
Regarding 3)
Thank you for this excellent remark. Although after prime immunization, AZ-primed individuals have significantly lower MFI for S1 and S2, the effect reverses with BNT boosting in heterologous AZ/BNT-boosted individuals, now showing significantly higher MFI for S1 and S2. We hypothesize that AZ priming might have induced a strong cellular response which is then stimulated by BNT boosting, resulting in the strong induction of antibodies with a single boost. This is also our explanation why AZ-primed individuals without exceeding threshold for anti-s1 IgG or neutralizing antibodies after priming dose can mount an antibody response as great as homologous BNT/BNT boosted individuals. A recent preprint of Liu et al. from the Com-COV Study Group that complements the initial reactogenicity data of Shaw et al., reports greatest induction of spike specific T cells for AZ/BNT individuals compared to AZ/AZ (lowest), BNT/BNT (second to lowest) and BNT/AZ (second to highest), supporting our hypothesis.
Regarding 4)
Neither Shaw et al. from the Com-COV study group nor Hillus et al. report differences in reactogenicity with regard to sex. We report in the limitations, that there is a female predominance in our study population which is why we did not perform a subgroup analysis on reactogenicity with regard to sex. Not all study participants reported on reactogenicity but we analyzed reactogenicity data on 53 BNT-primed (13 male, 40 female), 50 AZ-primed (16 male, 34 female), 53 homologous BNT/BNT-boosted (13 male, 40 female), 29 heterologous AZ/BNT-boosted (10 male, 19 female) and 14 homologous AZ/AZ boosted (4 male, 10 female). The female predominance in reactogenicity reporting matches with the female predominance of the whole study population. When looking at singular items of reactogenicity (overall reactogenicity, local event, systemic event) we do not find any significant differences in reactogenicity when comparing any event (local or systemic), any local or any systemic event in BNT-primed, AZ-primed and BNT/BNT boosted individuals (please see table). No statistics was performed for the homologous AZ/AZ and heterologous AZ/BNT group due to the small group size.
|
Group |
BNT |
AZ |
BNT/BNT |
AZ/AZ |
AZ/BNT |
|
Participants reporting |
53 |
50 |
53 |
14 |
29 |
|
Female N, (%) |
40 (75.47) |
34 (68) |
40 (75.47) |
10 (71.43) |
19 (65.52) |
|
Any event, local or systemic in females/males N, (%) |
23/9 (71.88) |
31/15 (67.39) |
33/11 (75) |
1/3 (25) |
11/10 (52.38) |
|
Local Event in females/males N, (%) |
17/9 (65.38) |
16/9 (64) |
19/9 (67.86) |
1/0 (100) |
7/8 (46.67) |
|
Systemic Event in females/males N, (%) |
13/5 (72.22) |
30/13 (69.77) |
30/10 (75) |
1/3 (25) |
7/8 (46.67) |
We agree that larger studies should look for differences in not only reactogenicity but also immunogenicity with regard to sex as we know that immunity is sex dependent. However, we did not perform this analysis as we focused on reactogenicity and immunogenicity with respect to different vaccination schemes and did find a female predominance in our study population right from the beginning.
Regarding 5)
We added the reference numbers accordingly.
Reviewer 3 Report
Article entitled “Heterologous ChAdOx1 nCoV-19/BNT162b2 prime-boost vaccination induces strong humoral responses among health care workers” is written well, and it is interesting. The following comments need to be addressed before considering further.
Comments :
- The entire theme is to study the “Heterologous” vaccination, but in this work, the numbers for Heterologous is very less, i.e., 35/166 (Ref: Table 01). The has been mentioned under limitation “(Line no 374)., the increasing the number may add more weightage to this study.
- Why this study mainly focused with “Health care workers” ?
- Also, I could find a few more articles with similar titles and similar claims, the author needs to check their data and justify the novelty of this work. Given below a few of those?
-
- Barros-Martins J, et al.Immune responses against SARS-CoV-2 variants after heterologous and homologous ChAdOx1 nCoV-19/BNT162b2 vaccination. Nat Med (2021). https://doi.org/10.1038/s41591-021-01449-9
- Groß et al., Heterologous ChAdOx1 nCoV-19 and BNT162b2 prime-boost vaccination elicits potent neutralizing antibody responses and T cell reactivity. medRxiv 2021.05.30.21257971; doi: https://doi.org/10.1101/2021.05.30.21257971
Author Response
Thank you for your valuable remarks and comments.
Regarding 1)
We agree that a higher number of heterologous vaccinees would have added to the power of our study. In our study, the boost group consists of 134 participants, whereof 82 (61.19%) received homologous BNT/BNT, 35 (25.93%) heterologous vaccination with AZ/BNT and 17 (12.69%) homologous AZ/AZ vaccination. Big study trials such as the Com-COV trial or the CombiVac S Study are looking at alternative vaccination strategies in bigger cohorts. We can perfectly contribute with our in-depth analysis of humoral responses to heterologous prime boost vaccination despite the small study size as our immunogenicity data is similar to study results from bigger cohorts published as pre-prints (Liu et al., Hillus et al., Groß et al.), implying correctness of our data despite smaller study population. Such independent studies with similar findings contribute to confidence in the results. Further, peer-reviewed data on heterologous vaccination remains scarce and is still greatly needed. A strength of our study is the in-depth characterization of humoral responses against different SARS-CoV-2 target antigens with a multiplex bead-based assay. Here, we see a stronger induction against S1 and S2 in the heterologous AZ/BNT boost group, which is why we hypothesize that AZ priming might have induced a strong cellular response which is then stimulated by BNT boosting. Recently pre-print published data on cellular immunity after COVID-19 vaccination by Liu et al. (Com-COV Study Group) supports our claim showing greatest induction of spike specific T cells in AZ/BNT vaccinated individuals.
Regarding 2)
Health care workers are at great risk for infection with COVID-19 which is why they were prioritized for vaccination in several countries. As vaccine rollout accelerated and severe, but rare cases of thromboembolic events after AZ priming were reported, vaccination strategy was adapted in several European countries despite limited data on safety and immunogenicity. Data regarding safety and immunogenicity in heterologous vaccination remains scarce and is therefore still greatly needed. Further, our data on safety and immunogenicity of different vaccination schemes in healthy individuals may serve as a reference population for other researchers investigating COVID-19 vaccines in other patient cohorts, including the sick and elderly.
Regarding 3)
We submitted our data on June 22nd, where the studies published by Barros-Martins et al. (published on July 14th) and Groß et al. (posted on a pre-print server July 15th) were not submitted yet. COVID-19 research is an emerging field issuing new results on a daily basis. We will include findings by Barros-Martins et al. and Groß et al. in our discussion as especially the findings of a heterologous vaccination scheme against emerging variants of concern (VOC) are of great value. Barros-Martins et al. and Groß et al. both tested on B.1.1.7 (Alpha), B.1.351 (Beta) and P.1 (Gamma), however, they did not test against the new emerging 1.617.2 (Delta) VOC. The results issued by Groß et al. are much smaller in sample size (26 heterologous AZ/BNT boosted individuals) compared to results published by Barros-Martins et al. (55 heterologous AZ/BNT vaccinees). We will further include recent pre-print published immunogenicity data by Liu et al. from the Com-COV Study and Hillus et al. who reported safety and immunogenicity data in 340 health care workers (whereof 51 with heterologous AZ/BNT vaccination).
Round 2
Reviewer 3 Report
The author has justified all comments